# Pandemic news information uncertainty— News dynamics mirror differential response strategies to COVID-19

Kristoffer Nielbo[1,2]*, Kenneth Enevoldsen[1], Rebekah Baglini[1,2], Elena Fano[3], Andreas Roepstorff[2], Jianbo Gao[4,5,6]

1 Center for Humanities Computing Aarhus, Aarhus University, Aarhus, Denmark, 2 Interacting Minds Centre, Aarhus University, Aarhus, Denmark, 3 National Library of Sweden, Stockholm, Sweden, 4 Center for Geodata and Analysis, Faculty of Geographical Science, Beijing Normal University, Beijing, China, 5 Institute of Automation, Chinese Academy of Sciences, Beijing, China, 6 International College, Guangxi University, Nanning, China

* kln@cas.au.dk

**Data Availability Statement:** Data cannot be shared publicly because of copyright by the newspaper publishing houses. Data are available from the Center for Humanities Computing Aarhus Institutional Data Access in UCloud (contact via

## Abstract

National differences in uncertainty, inequality, and trust have been accentuated by COVID-19. There are indications that the pandemic has impacted societies characterized by high uncertainty, inequality, and low trust harder than societies characterized by low uncertainty, equality, and high trust. This study investigates differential response strategies to COVID-19 as reflected in news media of two otherwise similar low uncertainty societies: Denmark and Sweden. The comparison is made using a recent approach to information dynamics in unstructured data. The main findings are that the news dynamics generally mirror public-health policies, capture fundamental socio-cultural variables related to uncertainty and trust, and may provide a measure of societal uncertainty. The findings can provide insights into evolutionary trajectories of decision-making under high uncertainty and, from a methodological level, be used to develop a media-based index of uncertainty and trust.

## Introduction

National outcomes of COVID-19 accentuate existing differences between societies characterized by different levels of uncertainty and trust [1]. On the one hand, societies characterized by high levels of economic and political uncertainty [2], as well as high economic inequality [3], and distrust in the government [4] (e.g., Brazil, Italy) have shown higher case fatality under COVID-19. On the other hand, societies with lower economic and political uncertainty and higher economic equality and trust in government (e.g., Finland, Switzerland) displayed lower case fatality. Interestingly, several nations that recently have moved towards higher levels of uncertainty (e.g., UK, US), also showed higher COVID-19 incidence rates [5, 6]. Because these differential pandemic outcomes correlate with various socio-cultural and economic variables, COVID-19 is well suited for natural experiments that investigate how nations' differential responses to COVID-19 may indicate a) possible evolutionary trajectories and b) non-obvious

chcaa@cas.au.dk) for researchers who meet the
criteria for access to confidential data.

**Funding:** This work was supported by
Carlsbergfondet with grant CF20-0044 (url: https://
www.carlsbergfondet.dk/). Kristoffer L. Nielbo
(KLN) was supported by Nordic eInfrastructure
Collaboration (NeIC) (url: https://neic.no//) and
Danish e-Infrastructure Cooperation's (DeIC) (url:
https://www.deic.dk/) with grant DeiC-AU1-L-
000001. The funders had no role in study design,
data collection and analysis, decision to publish, or
preparation of the manuscript.

**Competing interests:** The authors have declared
that no competing interests exist.

socio-cultural differences. One particularly interesting and underexplored contrast is the differential response to COVID-19 by two otherwise homogeneous societies with low uncertainty: Denmark and Sweden. Both nations belong to the Nordics group of universal welfare states; they show high levels of trust in the government, high economic equality, and low levels of economic and political uncertainty [7]. Denmark and Sweden generally share characteristics at institutional, systemic, and socio-cultural levels (e.g., media consumption) [8–10], yet they diverge significantly in response to and outcome of COVID-19. While Denmark was one of the first EU countries to initiate lockdown measures (March 13, 2020), Sweden avoided lockdown throughout COVID-19. Sweden predominantly used a *scientific response strategy* by responding to COVID-19 as a health issue to be managed by the public health agency [11]. Denmark combined a scientific response with a *political response strategy* that in addition, treated COVID-19 as a political issue managed by the government [1]. While Sweden's population is 2x larger than Denmark's, Sweden had 3.7x confirmed cases, and its case-fatality was 36% higher [12]. Importantly, we are only registering differences between Sweden and Denmark in response to and outcome of COVID-19, not arguing the specific causal scenario.

To properly understand differences in complex socio-cultural systems, like Sweden and Denmark, in response to COVID-19, we have to monitor and compare the states of both societies continuously. Ideally, we want to monitor all intrinsic variables related to uncertainty, inequality, and trust in each society. This strategy, however, is not feasible; instead, we suggest relying on the fundamental embedding theorem of chaos theory, which states that the detailed dynamics of a system that has an underlying attractor can be readily studied by reconstructing a suitable phase space of a scalar time series recorded from the system [13–15]. Chaos theory offers an elaborate scheme for generating aperiodic, highly irregular data from a deterministic system that can be characterized by only very few state variables instead of a random system with infinite numbers of degrees of freedom [16]. While the evolution of a complex social system may not be modeled as a dynamical system with a single attractor, we can assume that the dynamics of a large-scale social system can be approximated by switching between a large number of attractors, some of which may be simple, such as fixed points that may be associated with the dynamics of cultural information, while others may be complicated, including chaotic attractors [17, 18]. To understand uncertainty, inequality, and trust in the face of the pandemic, we need to find an adequate continuous variable related to cultural information shared by members of the societies during the event in question.

Several studies have shown that the associative structure of news media is highly sensitive to the socio-cultural dynamics, for instance, the rise and intricacies of modernity as reflected in historical newspapers [19, 20], and value-based differential response to negative events such as instability and war [21]. In a similar vein, it has been shown that the ordered one-dimensional representation of the word co-occurrence structure quite accurately captures historically relevant trends in newspapers [22]. By embedding the co-occurrence structure in a low-dimensional space, it has been shown that newspaper content reflects fundamental cultural movements and understandings from the 19th century onward [20] and that these context-dependent representations are sensitive to cultural bias as reflected in newspapers [23]. In continuation of the 'Culturomics' movement that used Google Books to show how lexical variation is sensitive to events [24], a wide range of studies has demonstrated that simple word and concept frequencies are sufficient for robust offline detection of major historical events [25] and can be used to model the evolution of complicated cultural processes such as the historical interdependencies between media and politics [26]. Fluctuations of time-dependent word frequencies have been shown to discriminate between classes of events that have class-specific fractal signatures, where the social-cultural class displays non-stationary and on-off intermittent behavior [27]. Even within the social-cultural class, different types of events (or stories

about events) seem to show fine-grained differences in their degree of self-affinity in newspapers [28].

In line with recent developments in information theory, studies have used information-theoretic models to track the states and dynamics of socio-cultural systems as reflected in lexical data [19, 29–32]. One paradigmatic study used relative entropy to study the development of Darwin's thinking in relation to his cultural context [29]. Both Shannon entropy and relative entropy have similarly been used in other studies to detect changes in prevalent mental states due to the socio-cultural context (e.g., state censorship, degree of recognition, religious observation) [31, 33]. One specific information-theoretic approach applies windowed relative entropy to dense low-dimensional text representations to generate signals that capture information *novelty* as a reliable content difference from the past and *resonance* as the degree to which future information conforms to said novelty [30, 32]. Taking a more dynamic perspective on this approach, one study has shown that discussion boards on social media where the novelty signal displays short-range correlations only and a particularly strong association with resonance are more likely to contain trending content [34]. Using the same approach, but combined with event detection, has also been shown to reliably predict significant change points in historical data [35].

On the specific intersection between news media, COVID-19, and uncertainty, a group of economists has developed an index for economic policy uncertainty based on dense probabilistic representations of newspaper articles [36]. The index correlates with existing market indices (e.g., VIX and BBO) and can accurately identify phase one of COVID-19 and other events associated with increased economic uncertainty. By using similar newspaper representations of front pages during COVID-19 and applying the above-mentioned information-theoretical approach, two studies have shown that the information dynamics of news during COVID-19 were reflective of societal and value-based responses to the pandemic and further argued that news media's response to the pandemic is a *decoupling* of the news content novelty and resonance (i.e., news information decoupling). The decoupling is particularly relevant in the context of socio-cultural responses to COVID-19 because it can be used to detect if and when a news media respond. Still, there are indications that the response may be mediated by alignment with the current government [37, 38].

In this paper, we propose to use the newspaper information dynamics as state indicators for Sweden and Denmark during the first phase of COVID-19. Specifically, we want to investigate if the novelty and resonance signal for national newspapers mirror known differences in strategic responses to COVID-19 (lockdown vs. no lockdown response and political vs. scientific strategy). Given the response difference and previous findings for Danish newspapers [37], we identify two possible, yet mutually exclusive predictions: a) *Mediated decoupling* according to which Denmark will show a decoupling between novelty and resonance and Sweden will not because the observed decoupling is mediated by response strategy, and b) *unmediated decoupling* according to which both nations will show a decoupling because the spread of COVID-19 alone is sufficient to result in a decoupling. Apart from comparing similar nations, the newspaper sample covers additional variables that are known to impact the information dynamics of newspapers, specifically political alignment (center-left vs. center-right) and newspaper type (tabloid vs. broadsheet/compact) [38].

## Materials and methods

### Data and normalization

The data set consists of all linguistic content (title and body text) from the front pages of five Danish newspapers (Berlingske, Ekstrabladet, Jyllands-Posten, Kristeligt Dagblad, and

**Table 1. COVID-19 first phase, Danish-Swedish newspaper data set.**

| Source | Country of origin | Type | Political alignment |
|---|---|---|---|
| Berlingske | *DK* | *C* | center-right |
| Ekstrabladet | *DK* | *T* | independent |
| Jyllands-Posten | *DK* | *C* | center-right |
| Kristeligt Dagblad | *DK* | *B* | independent evangelical |
| Politiken | *DK* | *B* | center-left |
| Aftonbladet | *SW* | *T* | center-left |
| Dagens Nyheter | *SW* | *C* | independent liberal |
| Expressen | *SW* | *T* | center-right |
| Svenska Dagbladet | *SW* | *B* | center-right |

Danish-Swedish newspaper data set. Column one contains the name of the newspaper, column two the newspaper's country of origin (*DK*: Denmark; *SW*: Sweden), three its type of newspaper (*B*roadsheet, *C*ompact, or *T*abloid), and four approximate political alignments of the newspaper. It is important to note that the newspapers do not have a direct affiliation with political parties today and that the alignment reflects their own classification. In some cases, this may not represent the perception of the readers.

Politiken) and four Swedish newspapers (Aftonbladet, Dagens Nyheter, Expressen, and Svenska Dagbladet), see Table 1. Front pages condense the most important news content and are, in comparison to full newspapers, more similar across conditions. Qualitatively similar results can be obtained from full newspapers, although the results are subject to considerably more variation. All newspapers in the sample are national and published daily, except Kristeligt Dagblad, which is only published six times per week from Monday to Saturday. Kristeligt Dagblad is kept in the sample because it is a national newspaper with substantial circulation and represents a specific Danish reader segment. All newspapers were sampled from December 1, 2019, to June 30, 2020. Importantly, internet access and, more generally, media consumption were approximately similar for both nations during the first half of 2020 [39, 40], which is typical for the Nordics group of universal welfare states. Furthermore, there were no major political events during the period that were not directly related to COVID [41].

To normalize linguistic content, numerals and highly frequent function words were removed, and the remaining data were casefolded and lemmatized using language-specific neural models trained on the Universal Dependencies Swedish LinES and Danish Dependency Treebank [42]. Subsequently, the data were represented as a bag-of-words (BoW) model using latent Dirichlet allocation to generate a dense low-rank representation of each front page. Parameter sweep was used for hyperparameter optimization and leave-p-out cross-validation was used for testing generalization. Note that with appropriate modifications to Eqs (4) and (5), the approach works for any probabilistic or geometric vector representation of documents. Novelty and resonance were estimated for windows of seven days, $w = 7$, representing the weekly news cycle.

## Novelty and resonance

Two related information signals were extracted from the temporally sorted BoW model: *Novelty* ($\mathcal{N}$) as an article $s^{(j)}$'s reliable difference from past articles $s^{(j-1)}, s^{(j-2)}, \ldots, s^{(j-w)}$ in window $w$:

$$\mathcal{N}_w(j) = \frac{1}{w} \sum_{d=1}^{w} JSD(s^{(j)} \mid s^{(j-d)}), \tag{1}$$

and *resonance* ($\mathcal{R}$) as the degree to which future articles $s^{(j+1)}$, $s^{(j+2)}$, ..., $s^{(j+w)}$ conforms to article $s^{(j)}$'s novelty:

$$\mathcal{R}_w(j) = \mathcal{N}_w(j) - \mathcal{T}_w(j) \tag{2}$$

where $\mathcal{T}$ is the *transience* of $s^{(j)}$:

$$\mathcal{T}_w(j) = \frac{1}{w}\sum_{d=1}^{w} JSD(s^{(j)} \mid s^{(j+d)}) \tag{3}$$

The novelty-resonance model was originally proposed in [30], but here we propose a symmetrized and smooth version by using the Jensen–Shannon divergence (*JSD*):

$$JSD(s^{(j)} \mid s^{(k)}) = \frac{1}{2}D(s^{(j)} \mid M) + \frac{1}{2}D(s^{(k)} \mid M) \tag{4}$$

with $M = \frac{1}{2}(s^{(j)} + s^{(k)})$ and $D$ is the Kullback-Leibler divergence:

$$D(s^{(j)} \mid s^{(k)}) = \sum_{i=1}^{K} s_i^{(j)} \times \log_2 \frac{s_i^{(j)}}{s_i^{(k)}} \tag{5}$$

## Change point detection

In accordance with [34], we model the media response to COVID-19 on novelty using a Bayesian approach to change point detection. For robustness, all results based on the change detection model were reproduced with a frequentist method for detection of change in the mean using the PELT algorithm. We assume that the time series contains two change points, $\tau_1$ and $\tau_2$, that can be located anywhere in the signal. This assumption is sufficient to detect at least one reliable decoupling (Decoupling Start and Decoupling End) in novelty. Aside from change points, the series is assumed to be stable and follow a normal distribution with varied mean, $\mu_i$, and singular variance, $\sigma$. This gives us the following model given the observed Novelty, $\mathcal{N}_i$:

$$\mathcal{N}_t = \begin{cases} \text{Normal}(\mu_1, \sigma) \text{ for } t < \tau_1 \\ \text{Normal}(\mu_2, \sigma) \text{ for } \tau_1 \leq t < \tau_2 \\ \text{Normal}(\mu_3, \sigma) \text{ for } t \geq \tau_2 \end{cases} \tag{6}$$

for which we wish to estimate the location of the change points $\tau_i$, means $\mu_i$ and variance $\sigma$, i.e. the following posterior:

$$P(\mu_i, \sigma, \tau_i | \mathcal{N}_t) = P(\mu_1, \mu_2, \mu_3, \sigma, \tau_1, \tau_2 | \mathcal{N}_t) \tag{7}$$

For estimation of the posterior, we used NUTS sampling with 4000 samples [43]. The estimation was done using slightly conservative priors assuming that the change points, $\tau_i$, can be anywhere in the sequence (with $\tau_2 > \tau_1$) and that the variance, $\sigma$, is stable across change points. Note that the half Cauchy prior distribution has a series of beneficial properties including its fat tail which allows for extreme values [44, 45]. These assumptions were modeled using the

following priors:

$$
\begin{aligned}
\mu_i &\sim \text{Normal}(0, 0.5) \\
\sigma &\sim \text{Half Cauchy}(0.5) \\
\tau_1 &\sim \text{Uniform}(0, \max(\mathbb{N}_t)) \\
\tau_2 &\sim \text{Uniform}(\tau_1, \max(\mathbb{N}_t))
\end{aligned}
\tag{8}
$$

### News information decoupling

Finally, in order to describe the information states before and after an event and confirm if a change point reflects a decoupling (i.e., that novelty decreases while resonance increases), we fit resonance on novelty to estimate the $\mathcal{N} \times \mathcal{R}$ slope $\beta_1$ before and during the event in question (e.g., the Danish lockdown):

$$
\mathcal{R}_i = \beta_0 + \beta_1 \mathcal{N}_i + \epsilon_i, \quad i = 1, \cdots, n.
\tag{9}
$$

where $\beta_0$ is the intercept and $\epsilon$ is a random variable representing the errors of the fit.

## Results

This section presents three results in order to compare Denmark and Sweden during the first phase of COVID-19. First, we establish a baseline for comparison of novelty and resonance before the first incidence of community transmission in Denmark and Sweden on March 9, 2020 (there is no official date for the first incidence of community transmission in Denmark, but by March 8-9 there was a nonlinear increase in the incidence rate, making it clear that the virus was circulating in the community, and could infect people with no history of travel. For methodological reasons, the date is therefore fixed between nations). Second, we test for reliable change points in the full novelty signal and compare the nation contrast. Finally, we inspect the $\mathcal{N} \times \mathcal{R}$ slope in order to determine if detected change-points reflect a decoupling, that is, and reliable $\mathcal{N} \times \mathcal{R}$ slope decrease. For *unmediated decoupling* to be valid, all newspapers, with the potential exception of tabloids, should display a decoupling at some point. In contrast, only Danish newspapers should show a decoupling that coincides with the lockdown for *mediated decoupling* to be valid.

### Novelty-resonance baseline

Fig 1 and Table 2 (column 2) show the $\mathcal{N} \times \mathcal{R}$ slopes from December 2019 to the first community transfer for Denmark and Sweden. Generally, a medium to a strong association between resonance and novelty can be observed for both countries: $\beta_M = 0.51$, $\beta_{SD} = 0.11$. Denmark ($\beta_M = 0.54$, $\beta_{SD} = 0.08$) and Sweden ($\beta_M = 0.48$, $\beta_{SD} = 0.13$). Tabloid newspapers (Ekstrabladet, Aftonbladet, Expressen) do not seem to deviate substantially from broadsheet and compact newspapers, which is supported by previous findings [38]. Nor does political alignment seem to impact the normal state of affairs systematically, but there is an indication that independent newspapers show a stronger association ($\beta_M = 0.68$, $\beta_{SD} = 0.01$) (c.f., Tables 1 and 2).

To summarize, under the normal state of affairs, novelty and resonance are coupled in the news media such that novel news items tend to resonate more than overused and repetitive items and vice versa. This finding confirms the intuition that news media, all things being equal, maintain their relevance by propagating news irrespective of newspaper type, political alignment, and nation.

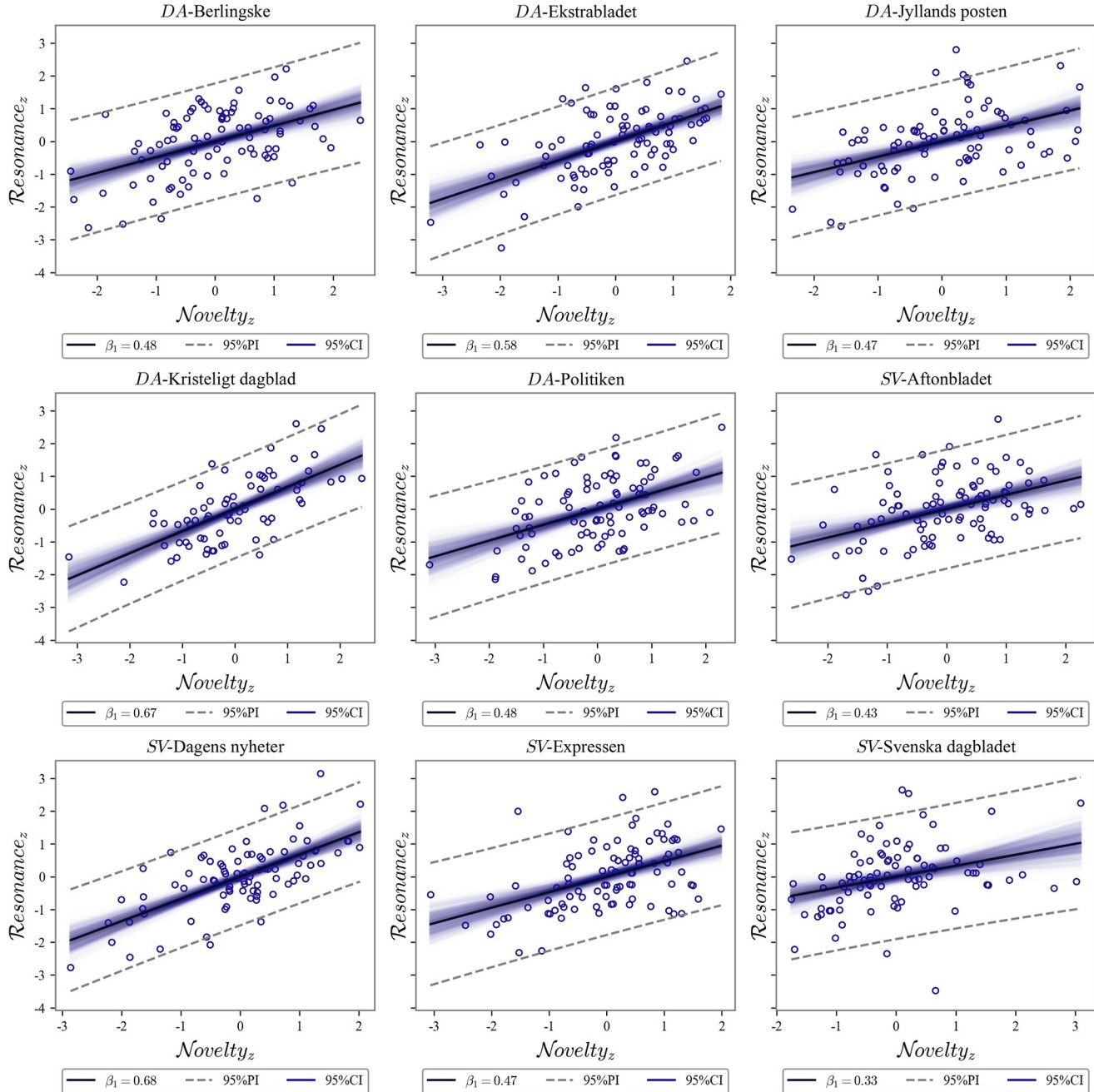

**Fig 1. Newspaper slopes.** Linear models of resonance on novelty (in standard scores) for inspection of $\mathcal{N} \times \mathcal{R}$ slopes, $\beta_1$, before the first incidence of community transmission in Denmark and Sweden. Punctured lines represent the 95% prediction intervals and transparent lines the (bootstrapped) 95% confidence intervals of the model.

## Novelty change points

There is an observable difference in novelty signals in the nation contrast, see Fig 2. All Danish newspapers, possibly with the exception of the tabloid newspaper Ekstrabladet, show a non-linear decrease in novelty that coincides with the lockdown (March 13 in week 11) and similarly a 'return to normal' around the re-opening (April 15 in week 16). Although the novelty

**Table 2. Baseline and lockdown newspaper slopes.**

| Source | Baseline | Lockdown |
|---|---|---|
| Berlingske | 0.48 [0.46, 0.5] | 0.26 [0.21, 0.32] |
| Ekstrabladet | 0.58 [0.57, 0.6] | 0.5 [0.45, 0.56] |
| Jyllands-Posten | 0.47 [0.45, 0.49] | 0.38 [0.33, 0.44] |
| Kristeligt Dagblad | 0.67 [0.65, 0.69] | 0.45 [0.37, 0.52] |
| Politiken | 0.48 [0.47, 0.49] | 0.29 [0.23, 0.34] |
| Aftonbladet | 0.43 [0.41, 0.45] | 0.55 [0.5, 0.61] |
| Dagens Nyheter | 0.68 [0.67, 0.7] | 0.61 [0.56, 0.65] |
| Expressen | 0.47 [0.46, 0.49] | 0.63 [0.58, 0.67] |
| Svenska Dagbladet | 0.33 [0.31, 0.36] | 0.33 [0.27, 0.39] |

$\mathcal{N} \times \mathcal{R}$ slopes, $\beta_1$, and 95% confidence intervals for *Baseline* (i.e., until the first incidence of community transmission in Denmark and Sweden) and the first Danish *Lockdown* (i.e. March 13 to April 15, 2020). The slope confidence intervals are estimated using bootstrapping from the original data set.

signals of Swedish newspapers are not constant during the period, they do not show coordinated and strong responses similar to that of Denmark. To validate the observable difference, we applied a change point detection model to the novelty signals, see Table 3. Be reminded that the model does not make any assumption about the location of a valid change point, see Eq 8. If, for instance, the sudden drop in Ekstrabladet in early May or the valley in Svenska Dagbladet starting in March represent reliable changes to the signal mean, then the model will confirm them irrespective of their location.

For Denmark, the first change point, 'Decoupling Start,' should separate pre-lockdown from lockdown centered on week 11, and the second, lockdown, 'Decoupling End', from post-opening (centered on week 16). Given that Sweden did not have a lockdown, any two valid change points are counted as True. For two change points to be valid, the interval estimates should be non-overlapping. The model confirms that all Danish broadsheet and compact newspapers, irrespective of political alignment, display two novelty change points that approximately coincide with the national lockdown; see Table 3. The first change point is placed in weeks 10-11; the second, however, is a matter of contention. The re-opening change point lies within April and displays up to a month's delayed response. Ekstrabladet is the only exception; while it does show a novelty drop during the lockdown, see Fig 2, a valid change point is not identified by the model. Swedish newspapers, except for Dagens Nyheter, do not show any reliable change points. Novelty in Dagens Nyheter does show a period of reliable change between weeks 3 and 11, which covers the period from the first incidence (January 31) to the first community transmission. Notice, however, from Fig 1 and Table 2 that the period is dominated by an increase in Novelty as opposed to the persistent decrease during the lockdown in Denmark.

Regarding front page content, what the change points in Danish newspapers indicate is content alignment around the lockdown. During the lockdown, all news became 'corona-news,' and terms associated with the national outbreak and crisis management (e.g., 'corona,' 'corona-virus,' 'lockdown,' 'cancellation') dominated the front pages, with titles such as 'Clear speech about Corona' [47], 'Storm of repatriations hits after corona' [48], and 'The government closes Denmark: now we must stand together,' [49] and seeped into all articles. Similar content alignment is not found in Swedish newspapers except for Dagens Nyheter. Compared to the Danish front pages, the Swedish content is more diverse, and mentions of terms associated with COVID-19 are sparse.

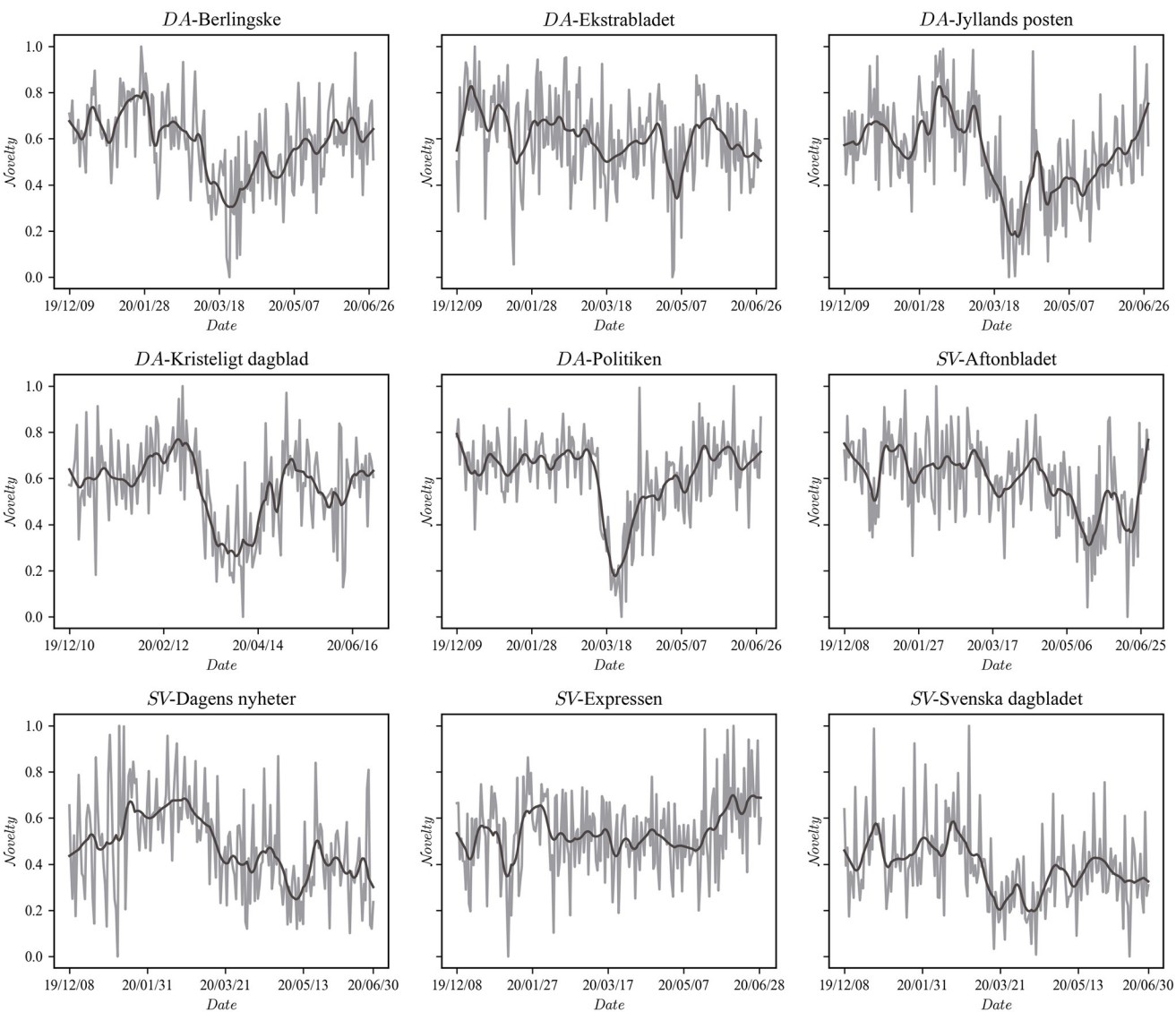

**Fig 2. Novelty signals.** The novelty for all newspapers (0 : 1 normalized) for comparison. The gray line shows the original signal and the black line is the smoothed signal using a nonlinear adaptive filter [33, 46]. Any time point represents the average uncertainty encoded in a front page given the preceding seven days, see Eq 1.

Returning to the inaccuracies in prediction for Danish newspapers. All valid initial change points, 'Decoupling Start', fall within a narrow window, less than a week from March 7 to match 13. By inspecting the front pages from this period, it becomes apparent that news items concerning COVID-19 were progressively taking over the front pages and that the media were becoming increasingly aware of the likelihood of the impending restriction policies. For the Danish newspapers, this thematic trend can be traced back to the first incident in Denmark. The location of the final change point, 'Decoupling End', is considerably more contested, placed somewhere in April. As with the initial change point, a gradual response is expected, but in reverse. While the 'official' re-opening was dated April 15, many restriction policies were still in effect (e.g., wage compensations, social distancing, assembly ban). An interesting contrast between political alignment can be observed from Table 4. Center-right newspapers

**Table 3. Newspaper change points.**

| Source | Decoupling Start | Decoupling End | Change Points |
|---|---|---|---|
| Berlingske | 20.03.07 [20.03.03, 20.03.09] | 20.04.28 [20.04.09, 20.05.08] | *True* |
| Ekstrabladet | 20.01.28 [20.01.02, 20.03.17] | 20.05.08 [20.01.16, 20.07.22] | *False* |
| Jyllands-Posten | 20.03.10 [20.03.08, 20.03.14] | 20.05.25 [20.05.21, 20.06.06] | *True* |
| Kristeligt Dagblad | 20.03.07 [20.03.05, 20.03.12] | 20.04.15 [20.04.11, 20.04.17] | *True* |
| Politiken | 20.03.13 [20.03.12, 20.03.13] | 20.04.08 [20.04.05, 20.04.08] | *True* |
| Aftonbladet | 20.04.21 [20.02.20, 20.05.15] | 20.06.12 [20.04.25, 20.06.27] | *False* |
| Dagens Nyheter | 20.01.13 [19.12.24, 20.01.19] | 20.03.13 [20.03.08, 20.03.18] | *True* |
| Expressen | 20.04.27 [20.01.06, 20.06.09] | 20.06.06 [20.05.21, 20.06.30] | *False* |
| Svenska Dagbladet | 20.03.07 [20.01.03, 20.03.16] | 20.04.17 [20.03.08, 20.04.30] | *False* |

Estimated temporal change points at 94% high density intervals for novelty. Column one contains the newspaper's name, columns two and three contains the beginning and end of a potentially reliable period of decoupling as represented in the newspaper, and the final column indicates if the specific source shows a valid set of change points. For two change points to be valid in the model, interval estimates must be non-overlapping.

(Berlingske and Jyllands Posten) are responsible for the 'inaccuracy' in estimates, by exhibiting a delayed linear novelty response during the post-opening. This may reflect their lack of support for the center-left government's opening strategy. In the aftermath of the lockdown, center-right newspapers focused more on the economic and societal consequences of the government's COVID-19 policies than their center-left counterpart (Politiken and, to some extent, Kristeligt Dagblad).

In summary, Denmark showed reliable change points coinciding with the national lockdown, while only one Swedish newspaper had reliable change points during the first phase. While this is a strong indicator that *mediated decoupling* is valid, and *unmediated decoupling* conversely invalid, event-based change in novelty is only half of the story. For an information decoupling to be present, resonance should (initially) increase during the event such that the medium to a strong association between novelty and resonance is momentarily weakened [37].

## Novelty-resonance decoupling

Following the procedure of [38], we only count $\mathcal{N} \times \mathcal{R}$ slopes relevant for decoupling if at least two reliable change points were identified with the change detection model. We show all linear fits of resonance on novelty in Fig 3 for the purpose of visual comparison and the slope values are available in Table 2. All Danish broadsheet/compact newspapers show novelty change points that coincided with the lockdown period, and they show a reliable decrease in

**Table 4. Newspaper change points.**

| Source | $\mathcal{N}_{pre}$ | $\mathcal{N}_{during}$ | $\mathcal{N}_{post}$ |
|---|---|---|---|
| Berlingske | 0.36 [0.35, 0.37] | 0.29 [0.27, 0.31] | 0.34 [0.34, 0.35] |
| Jyllands-Posten | 0.29 [0.28, 0.3] | 0.23 [0.22, 0.24] | 0.27 [0.26, 0.28] |
| Kristligt Dagblad | 0.27 [0.26, 0.28] | 0.19 [0.18, 0.21] | 0.26 [0.25, 0.27] |
| Politiken | 0.27 [0.26, 0.28] | 0.15 [0.14, 0.17] | 0.26 [0.25, 0.26] |
| Dagens Nyheter | 0.21 [0.19, 0.23] | 0.25 [0.24, 0.27] | 0.19 [0.18, 0.2] |

Novelty values at 94% high density intervals before during and after the change points for the four broadsheet newspapers that showed reliable points, see Table 3.

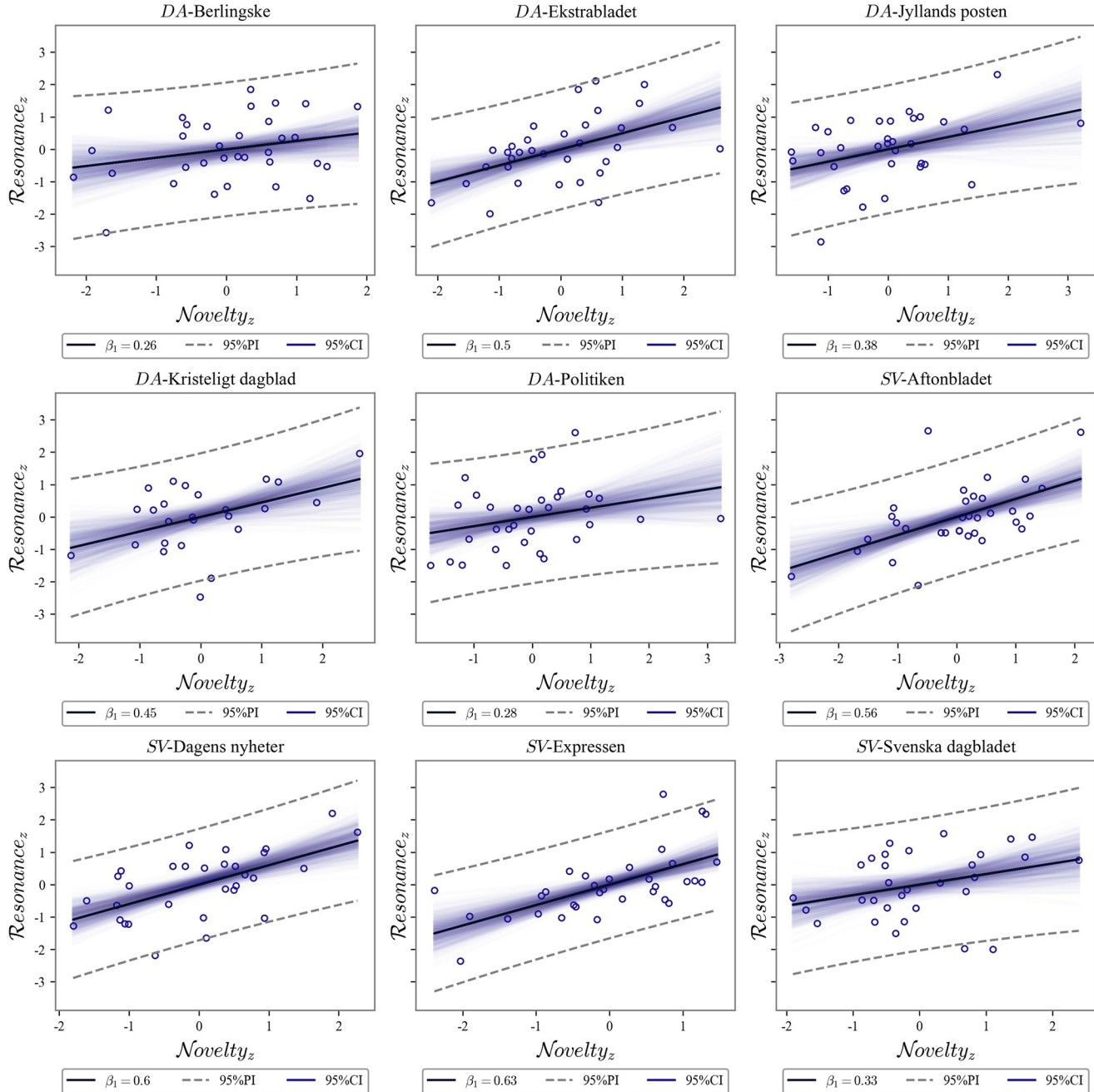

**Fig 3. Lockdown newspaper slopes.** Linear models of resonance on novelty (in standard scores) during the Danish lockdown period with the exception of Dagens Nyheter (see text). Swedish (Aftonbladet, Expressen, and Svenska Dagbladet) and Danish (Ekstrabladet) newspapers that do not show valid change points are provided for comparison only. Punctured lines represent the 95% prediction intervals and transparent lines the (bootstrapped) 95% confidence intervals of the model. Notice how Danish newspapers that show valid change points (Berlingske, Jyllands Posten, Kristeligt Dagblad and Politiken) also display a considerable drop in the $\mathcal{N} \times \mathcal{R}$ slope, c.f., Fig 1.

the novelty resonance association, from $\beta_M = 0.53$, $\beta_{std} = 0.8$ to $\beta_M = 0.34$, $\beta_{std} = 0.8$, that is an average relative decrease of 44%. In contrast, similar Swedish newspapers only showed an 8% decrease in the same period. Looking at newspaper type, tabloids show an average growth of 13% in the $\beta$ coefficient during the specified period. Ekstrabladet does show a slope decrease

during the period, which can also be observed in Table 2. The shift in Ekstrabladet's average novelty is, however insufficient evidence of a change point. It could be argued that the tabloid paper does show a tendency in a direction similar to other Danish newspapers, but it should be noted that [38] did not find a similar tendency for the other national Danish tabloid BT.

For the Swedish broadsheet newspapers, only Dagens Nyheter shows evidence of decoupling with a 12% decrease in the $\beta$ coefficient, which, although not on the same scale as the Danish response, does conform to the decoupling hypothesis. The remaining Swedish newspapers either show no change in novelty or an increase during the Danish lockdown, see Table 2. Notice that the Danish lockdown period coincides with key events related to the pandemic (e.g., public health advice to maintain distance, use disinfectant, and work from home, and the peak in Deaths for the first phase of COVID-19) [11].

To further explore the findings, we inspect average changes in novelty before, between, and after the change points in Table 4. It can be observed that all Danish show a statistically reliable decrease in novelty ($MD = 5.75$) when transitioning to the lockdown state and and reliable mean increase when reopening ($MD = 3.75$). Although Dagens Nyheter's change points loosely overlap with the early COVID-19 breakout in January and end with the first community incidence in March, it shows an increase in novelty during the period, and conversely a relative decrease in resonance. This reflects an influx of new information related to COVID-19 that does not stabilize during the period. While this behavior can, at least in the limit, be classified as a decoupling of information, it moves in the opposite direction of the predicted decoupling. It is important to reiterate that in the case of Dagens Nyheter, the slope decrease is also considerably smaller.

To summarize, only Danish broadsheet newspapers show news information decoupling as defined in [37], which lends support to the *mediated decoupling* prediction and, conversely, rules out *unmediated decoupling*. In other words, with the exception of tabloids, Danish newspapers show a decoupling of information during the first phase of COVID-19 and Swedish newspapers do not. This Danish news information decoupling more or less coincides with the lockdown and, finally, the interval uncertainty may be attributable to political alignment.

## Discussion

This paper has shown how a bifurcation in national response strategies to COVID-19 is mirrored in the information dynamics in news media. Sweden used a predominantly scientific response strategy and avoided lockdown during COVID-19, which was mirrored in a lack of change points in the news content and, in some cases, a slight increase in novelty due to new information regarding COVID-19 and possibly other news events. In contrast, Denmark combined a scientific with a political response strategy resulting in one of the earliest international lockdowns. This response is mirrored in the information decoupling of content novelty from resonance. These findings are particularly interesting because the compared nations (i.e., Denmark and Sweden) are very similar at institutional, systemic, and socio-cultural levels. Danish news content showed a highly coordinated response with the state lockdown (i.e., news information decoupling) [41], while Swedish news dynamics did not change significantly during the pandemic's first phase mirroring a less coordinated response by the Swedish government [11].

One way to interpret these differences in media behavior is as a state indicator. During the first phase of COVID-19, Sweden's scientific response gravitated towards a higher entropy state, while Denmark's combined scientific and political response lowered its entropy state. Information dynamics (i.e., novelty and resonance dependencies) do to some extent reflect societal uncertainty and can therefore serve as an index of a nation's general

state of information entropy. A news information decoupling, that is, when novelty decreases and resonance increases, reflects a reduction of uncertainty possibly in response to otherwise unpredictable and negative events [38]. This study has however shown that decoupling is *mediated* by political decision-making and not a necessary media response to catastrophes. It could be argued that a coordinated response, the possibility of, depends on the state of information entropy in a given society and that decoupling is only possible in low entropy states—thereby assuming that Sweden was already in a high entropy state when the pandemic spread. This interpretation does however ignore that factors like suppression and proscription of speech and writing (e.g., media censorship) can facilitate a highly coordinated media response in high entropy societies. China, for instance, has expelled several foreign journalists that did not support the official view of the pandemic's economic impact [50].

What then, can explain the observed response difference to COVID-19 between Denmark and Sweden? One proposal is a 'throw-of-the-dice' explanation, according to which Denmark and Sweden generally follow similar evolutionary trajectories, but, due to random fluctuation, diverged at a point in a time characterized by catastrophic uncertainty [51]. COVID-19 did arrive earlier in Sweden than Denmark and the Swedish constitution complicates lockdown in other conditions than war. Both of these factors could have 'weighted' the dice and pushed Sweden towards its current trajectory. An alternative account is 'uncertainty-trust mediation', that is, differences in trust and uncertainty between Denmark and Sweden may have predated COVID-19. A recent survey study has shown that Danes did indeed 'rally around the flag' during COVID-19 [41] and more generally showed a higher degree of trust in the government and health authorities during the pandemic compared to Sweden, and, finally, that trust was politicized from the beginning of COVID-19 in Sweden [7]. The same study also found that even before 2020, trust in the government was significantly lower in Sweden than in Denmark. It is likely that differences in trust have contributed to higher levels of uncertainty pushing Sweden toward a higher entropy state. Several psychological studies have shown that trust, or the lack of it, is a significant driver of uncertainty [52]. What could further support this account is the higher level of COVID-19 vaccine hesitancy in Sweden than in Denmark [53]. On this account, we would still predict that societies in low entropy states will migrate toward a coordinated response (like Denmark), and societies in high entropy states toward an uncoordinated response (like Sweden). The question of generalizability is beyond the scope of this paper, but a recent study has identified similar signatures in Dutch newspapers with an information-theoretic model for event description over a long time span [54]. From a methodological perspective, news information dynamics can provide a valuable tool for media-based indices that can supplement existing economic and policy-based indices of uncertainty and trust [36] at least in some national contexts.

Several observations have been made within conditions. In the Danish sample, although the Ekstrabladet did show a tendency, we could not confirm a coordinated response from the tabloid during the first phase of COVID-19. This finding has previously been shown for major tabloids in Denmark [38]. The same study argued that differences in post-lockdown behavior reflect political alignment and that because the Danish government during the lockdown was center-left, the center-right newspapers were more skeptical of the government's implementation of an opening than the center-left. The distinct patterns of variation within and between conditions are likely to contain useful information about news media. Further comparisons of left vs. right-wing newspapers, tabloid vs. broadsheet newspapers, silly season, and other seasonal effects, offer interesting venues for media and journalism researchers.

## Conclusion

This study has shown how the dynamics of probabilistic newspaper representations can be used to detect and discriminate different response strategies to COVID-19 in two otherwise very similar Nordic countries. One nation in question, Denmark, showed an information decoupling in national news media that coincided with the first phase of COVID-19 and particularly the lockdown. In contrast, the propagation of news continued as normal in Sweden during the same period. We offer two accounts of observed differences in news information and pandemic response strategies, one according to which they are the results of random fluctuations, and, an alternative, where the differences may be reflective of underlying socio-cultural differences associated with trust and uncertainty. From a methodological perspective, the study provides the foundations of a tool for media-based change detection and characterization of societal uncertainty.

## Acknowledgments

The authors would like to thank Berlingske Media, Infomedia, JP/Politikens Hus, Kristeligt Dagblad, Aftonbladet, Dagens Nyheter, Expressen, and Svenska Dagblade for access to proprietary data.

## Author Contributions

**Conceptualization:** Kristoffer Nielbo, Rebekah Baglini, Andreas Roepstorff, Jianbo Gao.

**Data curation:** Elena Fano.

**Formal analysis:** Kristoffer Nielbo, Kenneth Enevoldsen.

**Methodology:** Kristoffer Nielbo, Rebekah Baglini.

**Resources:** Andreas Roepstorff.

**Software:** Kristoffer Nielbo, Kenneth Enevoldsen.

**Supervision:** Rebekah Baglini.

**Visualization:** Kristoffer Nielbo.

**Writing – original draft:** Kristoffer Nielbo.

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
