## [Decision Letter · Decision Letter 0]

22 Apr 2022

PONE-D-21-33093Pandemic news information uncertainty: News dynamics mirror differential response strategies to COVID-19PLOS ONE

Dear Dr. Nielbo,

Thank you for submitting your manuscript to PLOS ONE. After careful consideration, we feel that it has merit but does not fully meet PLOS ONE’s publication criteria as it currently stands. Therefore, we invite you to submit a revised version of the manuscript that addresses the points raised during the review process.

The paper is well written however need to provide evidence providing justification of claim. For example, how responses effects dynamic behaviour of the newspaper. The reviewer 2 raised several significant concerns those need to be fixed before arriving final decision.

We look forward to receiving your revised manuscript.

Kind regards,

Siuly Siuly, PhD

Academic Editor

PLOS ONE

Journal Requirements:

3. Thank you for stating the following in the Acknowledgments Section of your manuscript: "This research was supported the “HOPE - How Democracies Cope with COVID-19”-project funded by The Carlsberg Foundation with grant CF20-0044, NeiC’s 

Nordic Digital Humanities Laboratory project, and DeiC Type-1 HPC with project

DeiC-AU1-L-000001."

Please remove any funding-related text from the manuscript and let us know how you would like to update your Funding Statement. Currently, your Funding Statement reads as follows: "The funders had no role in study design, data collection and analysis, decision to publish, or preparation of the manuscript."

Reviewers' comments:

Reviewer's Responses to Questions

**Comments to the Author**

1. Is the manuscript technically sound, and do the data support the conclusions?

Reviewer #1: Yes

Reviewer #2: Partly

2. Has the statistical analysis been performed appropriately and rigorously? 

Reviewer #1: I Don't Know

Reviewer #2: No

3. Have the authors made all data underlying the findings in their manuscript fully available?

Reviewer #1: Yes

Reviewer #2: No

4. Is the manuscript presented in an intelligible fashion and written in standard English?

Reviewer #1: Yes

Reviewer #2: Yes

5. Review Comments to the Author

Reviewer #1: The paper was nicely organised and presented in a very well structured way. It was very unique works and very logical during pandemic. Methods and analysis parts are well arranged. But one fact is that, is this generalized for all over the country in the world??

Reviewer #2: 1. According to the title, the manuscript should provide evidence supporting how different response strategies to the COVID-19 epidemic influence the dynamic change of newspapers. The only evidence was based on the timeline of disease progression, which was not solid enough in the modern society of diversion. There were many possible confounding factors, such as political events, international pandemics, newspaper selling, internet resources, and others. In addition, the manuscript failed to provide a detailed discussion about the slightly different starting and ending points of other different news sources in the same country (Table 2).

2. Further clarification of some statistical explanations is required, for example, in the Novelty-Resonance Baseline. The authors stated Denmark shows a stronger average association than Sweden. However, the values of mean and SD overlapped for the two countries. In Fig. 2, some "significant" decreases in novelty for Sweden newspapers were not pointed out. The date of the observed "valleys" was different from any given local COVID-19 epidemic events, which may indicate the dynamic change of newspaper were not merely driven by COVID-19, but for many other reasons. The difference between Fig.1 and Fig.3 should be addressed by mathematical analyses.

3. Some data was protected due to copyright. However, it is interesting to know some words used in BoW models. Adding some examples of so-called "novelty" and "resonance" of newspaper headlines or stories would also increase the readability of this study.

4. For a newspaper, the composition of readers is as important as political alignment; it is worth surveying the population difference of newspaper readers. For example, urban and rural residents may focus differently on the COVID-19 and national response.

5. Overall, a lot of description of the manuscript is subjective.

6. PLOS authors have the option to publish the peer review history of their article (what does this mean?). If published, this will include your full peer review and any attached files.

Reviewer #1: No

Reviewer #2: No

---

## [Author Response · Author response to Decision Letter 0]

8 Aug 2022

# Response to Reviewers #

__Title__: Pandemic news information uncertainty - News dynamics mirror differential response strategies to COVID-19

__Authors__: K. Nielbo, K. Enevoldsen, R. Baghlini, E. Fano, A.Roepstorff, J.Gao

__Contact__: Kristoffer Nielbo, kln_at_cas_dot_au_dot_dk

We would like to use the opportunity to thank the reviewers for their valuable comments. We have done our best to address their concerns and improve the manuscript accordingly. Below we will summarize their comments in sub-headings and reference where we have addressed them in the revised manuscript.

The manuscript has been through an additional language review after the revisions.

## R1: Generalizability of the results ##

The short answer is that we do not know how well at the moment this generalizes, but we have added references to studies that point in that direction and discuss the methodological relevance of our approach. We have added additional content regarding the scope of the study, see additional lines: 94-97 , 350-355.

## R2: Evidence, confounding factors and variation in start/end points ##

In general we have made it more explicit that the manuscript is only modeling how newspapers _mirror_ two response strategies (scientific vs. scientific & political) and the related behavior (lockdown vs no lockdown), see additional lines: 23-27; 94-97; 101; 105-108; 300-306

This comment has multiple sub-items:

1. factors: political events, international pandemics, newspaper selling, internet resources: 

 * The motivation for choosing Sweden and Denmark was exactly that both nations are very similar on most confounding factors. To address the specific concerns we have reference reports in footnote four that show that internet access and media consumption during 2020 were very similar for Denmark and Sweden. Political events are more complicated, but from a qualitative survey of news, we have any major political events during the period that were not directly related to COVID-19 (also mentioned in footnote four).

 * We mention key events in Sweden that co-insides with the Danish lockdown, lines 272-278.

2. start/end variation

 * Regarding the slightly different starting and ending points in Denmark, we have added a explanation essentially about the gradual nature of the COVID-19 news event, see lines 230-244 

## R2: Clarification of statistical explanations ##

This comment has multiple sub-items:

1. Novelty-Resonance Baseline

 * For more detail on the baseline, we have now added table 2 that contains slopes and 95% confidence intervals for figure 1.

2. The authors stated Denmark shows a stronger average association than Sweden. However, the values of mean and SD overlapped for the two countries

 * deleted

3. "significant" decreases in novelty for Sweden newspapers were not pointed out.

 * We checked the robustness of the model's detected 'valleys' (as opposed to other observable `valleys') with a frequentist method for change detection, now mentioned in footnote 5.

 * Lines 196-203 explain in more detail that the model is agnostic to the location of the change points, iow. if 'the decreases in novelty for Sweden' were indeed significant changes to the mean, the model will detect them.

 * Additional numerical explanation of the change in newspapers with change points in lines 280-82 

4. difference between Fig.1 and Fig.3 should be addressed by mathematical analyses.

 * For analysis of Fig.1 vs Fig.3, we have now added table 2 that contains slopes and 95% confidence intervals for both.

 * Additional explanation about the difference between the figures in lines 272-278.

## R2: Provide content examples ##

While all newspaper content is copyrighted and we are not allowed to share parts in any form (except for reviewers), we have provided a few examples of titles (from a public media database) and an explanation of changes to content in lines 221-230.

## R2: Description in the ms is subjective ##

We have added additional references to recent studies that support the more descriptive parts

* Baekgaard M , Christensen J, Madsen JK , Mikkelsen KS Rallying around the

flag in times of COVID-19 Journal of Behavioral Public Administration.

2020;3(2). doi:10.30636/jbpa.32.17

* Brusselaers N, Steadson D, Bjorklund K, Breland S, Stilhoff S ¨orensen J, Ewing A,

Bergmann S, Steineck G. Evaluation of science advice during the COVID-19

pandemic in Sweden Humanities and Social Sciences Communications.

2022;9(1):1–19. doi:10.1057/s41599-022-01097-5

* Wevers M, Kostkan J, Nielbo KL Event Flow - How Events Shaped the Flow of

the News, 1950-1995 Proceedings for the CHR 2021: Computational Humanities

Research Conference, November 17–19, 2021.

July 6, 2022 18/18

---

## [Decision Letter · Decision Letter 1]

10 Nov 2022

Pandemic news information uncertainty: News dynamics mirror differential response strategies to COVID-19

PONE-D-21-33093R1

Dear Dr. Nielbo,

We’re pleased to inform you that your manuscript has been judged scientifically suitable for publication and will be formally accepted for publication once it meets all outstanding technical requirements.

Kind regards,

Tianyang Liu

Academic Editor

PLOS ONE

Additional Editor Comments (optional):

Reviewers' comments:

Reviewer's Responses to Questions

**Comments to the Author**

1. If the authors have adequately addressed your comments raised in a previous round of review and you feel that this manuscript is now acceptable for publication, you may indicate that here to bypass the “Comments to the Author” section, enter your conflict of interest statement in the “Confidential to Editor” section, and submit your "Accept" recommendation.

Reviewer #2: All comments have been addressed

2. Is the manuscript technically sound, and do the data support the conclusions?

Reviewer #2: Yes

3. Has the statistical analysis been performed appropriately and rigorously? 

Reviewer #2: Yes

4. Have the authors made all data underlying the findings in their manuscript fully available?

Reviewer #2: Yes

5. Is the manuscript presented in an intelligible fashion and written in standard English?

Reviewer #2: Yes

6. Review Comments to the Author

Reviewer #2: The revised version of the manuscript has responded well to the previous recommendations. The global generalization of the conclusion may not be impossible; however not within the scope of the current study.

7. PLOS authors have the option to publish the peer review history of their article (what does this mean?). If published, this will include your full peer review and any attached files.

Reviewer #2: **Yes: **Sheng-Yang Huang

---

## [Editor Report · Acceptance letter]

2 Jan 2023

PONE-D-21-33093R1 

Pandemic news information uncertainty - News dynamics mirror differential response strategies to COVID-19 

Dear Dr. Nielbo:

I'm pleased to inform you that your manuscript has been deemed suitable for publication in PLOS ONE. Congratulations! Your manuscript is now with our production department. 

Kind regards, 

on behalf of

Professor Tianyang Liu 

Academic Editor

PLOS ONE